# Anti-Aminoacyl Transfer-RNA-Synthetases (Anti-tRNA) Autoantibodies Associated with Interstitial Lung Disease: Pulmonary Disease Progression has a Persistent Elevation of the Th17 Cytokine Profile

**DOI:** 10.3390/jcm9051356

**Published:** 2020-05-06

**Authors:** Espiridión Ramos-Martinez, Ramcés Falfán-Valencia, Gloria Pérez-Rubio, Mayra Mejia, Ivette Buendía-Roldán, Montserrat I. González-Pérez, Heidegger N. Mateos-Toledo, Jorge Rojas Serrano

**Affiliations:** 1Experimental Medicine Research Unit, Facultad de Medicina, Universidad Nacional Autónoma de México, Mexico City 06720, Mexico; espiri77mx@yahoo.com.mx; 2HLA Laboratory, Instituto Nacional de Enfermedades Respiratorias Ismael Cosío Villegas, Mexico City 14080, Mexico; rfalfanv@iner.gob.mx (R.F.-V.); glofos@yahoo.com.mx (G.P.-R.); 3Interstitial Lung Disease and Rheumatology Unit, Instituto Nacional de Enfermedades Respiratorias Ismael Cosío Villegas, Mexico City 14080, Mexico; medithmejia1965@gmail.com (M.M.); ixchelglez19@gmail.com (M.I.G.-P.); dr_heidegger@msn.com (H.N.M.-T.); 4Translational Research Laboratory on Aging and Pulmonary Fibrosis, Instituto Nacional de Enfermedades Respiratorias Ismael Cosío Villegas, Mexico City 14080, Mexico; ivettebu@yahoo.com.mx; 5Profesor, Programa de Maestría y Doctorado en Ciencias Médicas, Facultad de Medicina, Universidad Nacional Autónoma de México, Mexico City 14080, Mexico

**Keywords:** anti-synthetase syndrome, interstitial lung disease, inflammatory cytokine profile, IL-17A, Th17

## Abstract

Anti-tRNA autoantibodies are associated with interstitial lung disease (ILD), in at least two clinical scenarios: the anti-synthetase syndrome (ASSD) and interstitial pneumonia with autoimmune features (IPAF). Under pathological conditions, cytokines indicate the participating elements and the course of inflammatory phenomena. We aimed to quantify serum concentrations of different inflammatory cytokines profiles in patients with anti-tRNA associated ILD (anti-tRNA-ILD) and estimate the association between these and ILD improvement and progression. Serum levels of 18 cytokines from baseline and after six months of treatment of ILD patients’ positives to anti-tRNA were included in the current study. At six months, patients were classified as with or without ILD progression. A total of 39 patients were included (10 anti-Jo1, eight anti-PL7, 11 anti-PL12, and 10 anti-Ej). Three patients (7.6%) had ILD progression (progressors patients, PP) and showed statistically higher levels in IL-4, IL-10, IL-17A, IL-22, GM-CSF, IL-1β, IL-6, IL-12, IL-18, and TNF-α, compared to patients without disease progression (no progressors patients, NPP). IL-17A, IL-1β, and IL-6 (T-helper-lymphocyte (Th)17 inflammatory cytokine profile) were elevated and had a high discriminatory capacity in distinguishing ILD PP of those NPP at follow-up. Overall, there is an association between the cytokines of the Th17 inflammatory profile and the ASSD progression.

## 1. Introduction

The anti-aminoacyl transfer-RNA-synthetases (anti-tRNA) autoantibodies are diverse and include anti-EJ (anti-glycyl), anti-OJ (anti-isoleucyl), anti-PL7 (anti-threonyl), anti-PL12 (anti-alanyl), anti-SC (anti-lysil), anti-KS (anti-asparaginyl), anti-JS (anti-glutaminyl), anti-Ha or anti-YRS (anti-threonyl), anti-tryptophanyl, and anti-Zo (anti-phenylalanyl) and anti-Jo1 (anti-histidyl) [1], the latter being the most frequent. The anti-tRNA are associated to interstitial lung disease ILD (anti-tRNA-ILD) in at least two clinical scenarios: the antisynthetase syndrome (ASSD), an inflammatory myopathy (IM) characterized by arthritis, mechanic’s hands, fever, Raynaud’s phenomenon, and ILD [1], as well as interstitial pneumonia with autoimmune features (IPAF) [2] (a classification criterion for ILD patients and features of autoimmunity, but not definitive for established criteria of connective tissue disease). Both ASSD and IPAF positive to anti-tRNA share the same ILD high-resolution tomographic patterns (HRCT) and response to medical treatment [3,4], thus it seems that both clinical conditions are clinical variations of the same pathophysiological process. The anti-tRNA-ILD treatment is centered on the administration of corticosteroids and a wide variety of immunosuppressive drugs; however, the effectiveness of the treatment depends on conditions not yet fully clarified [5].

The first events in the IM pathogenesis take place in the lung [6], causing distress and cell death, that eventually, produce antigen exposure that usually would not have close contact with the immune system [7]. Moreover, some genetic background would favor the loss of self-tolerance, which would trigger an abnormal sensitization towards own cells and tissues turning a local event into a systemic disease following pathways, however this is still poorly understood [7].

The immune system was defined as a protective factor during infectious diseases over a century ago [8]. Nonetheless, modern knowledge reveals that its primary function is the maintenance of tissue homeostasis and integrity of the system as a consequence of the recognition of the self and the non-self, with subsequent immunological elimination of potential risks [9]. Defective discrimination between the self and the non-self culminates in the development of autoimmune pathologies such as anti-tRNA-ILD [10]. Despite the efforts of different research groups to date, no specific therapeutic biomarkers or targets have been proposed, which would favor a better understanding of the syndrome and patient management. The different cytokine profiles produced by T-helper-lymphocytes (Th1, Th2, Th9, Th17, etc.) could be possible therapeutic targets and guides to know the inflammatory pathways involved. This work aimed to quantify cytokines of different inflammatory profiles in ILD patients positive to anti-tRNA, at baseline clinical evaluation before medical treatment for ILD, and at six months after the initiation of ILD treatment, to establish relationships between the concentrations of these mediators and clinical evolution of ILD positive to anti-tRNA.

## 2. Material and Methods

### 2.1. Patients

Participants were enrolled between January 2017 and June 2019. To be included in this study, patients must have had the diagnosis of ILD confirmed with high-resolution chest tomography (HRCT) and be positive to one of the following autoantibodies: anti-Jo1, anti-PL7, anti-PL12, or anti-Ej. We invited to participate in the study, patients with different anti-tRNA, to have as possible, a similar number of patients of the different anti-tRNA analyzed. Patients from all over Mexico are referred to the Interstitial Lung Disease and Rheumatology Unit (ILD&RU), at the Instituto Nacional de Enfermedades Respiratorias, Ismael Cosio Villegas (INER), a national referral center for respiratory diseases; patients referred to the ILD&RU are evaluated by a multidisciplinary group (pulmonologists, radiologists, and a rheumatologist). All patients were managed with the same medical treatment: an initial high dose of prednisone (0.5–1 mg/kg/ daily single morning dose) for six weeks, followed by a tapering dose of prednisone, combined with methotrexate 12.5–25 mg per week, plus leflunomide 20 mg per oral per day. All patients received folic acid and vitamin D supplementation therapy [4].

At baseline, we registered the duration of pulmonary symptoms (dyspnea and cough) pulmonary function tests (PFTs), which included the diffusing capacity for carbon monoxide (DL_CO_), spirometry, and plethysmography. Also, baseline serum creatinine kinase (CK) levels were recorded, as well as the history of proximal muscle weakness, Raynaud’s phenomenon, sclerodactyly, dermatomyositis rash, proximal dysphagia, modified Borg dyspnea scale and smoking history. After six months of follow-up, PFTs were performed to evaluate the evolution of ILD and to classify patients as with ILD progression, or with treatment response or stability of lung disease. Disease progression and treatment response on PFTs were defined as a decrease or increase in forced vital capacity (FVC) by more or less than 10% of those predicted, respectively, and/or a reduction or increase in DL_CO_ by more or less than 15% of predicted, respectively, similar to the established criteria for idiopathic pulmonary fibrosis. The local institutional review board approved the study protocol (approval code number: C23-17). Informed consent was given to all patients to participate in the study.

### 2.2. Pulmonary Function Tests

PFTs were performed in the Department of Respiratory Physiology, of the INER, a specialized respiratory physiology laboratory. In every measurement of PFTs, weight and standing height were measured by a digital scale (models 206 and 769, Seca, Hamburg, Germany). Spirometry (to obtain forced vital capacity) and DL_CO_ were performed using the commercial equipment Easy One Pro and Easy One Pro Lab (Ndd® Zurich, Switzerland). The data were expressed as percentages of the predicted values. The predicted values for each subject, according to sex, age, height, and weight, were obtained from the PLATINO study [11] and National Health and Nutrition Examination Survey (NHANES) tables [12] studies. All spirometry and DL_CO_ tests fulfilled the acceptability and reproducibility criteria of American Thoracic Society / European Respiratory Society (ATS/ERS 2005) [13,14].

### 2.3. High-Resolution Computed Tomography (HRCT) Evaluation

HRCT was performed at baseline evaluation with a 1.0 or 1.5 mm thick axial section taken at 1 cm intervals and reconstructed using a high spatial frequency algorithm. Between 20 and 25 CT, scan images were acquired for each patient. HRCT was blind evaluated by two experts (M.M. and H.N.M.-T.). Experts assessed the HRCT and classified the images according to the official ATS/ERS Statement of the International Multidisciplinary Classification of the Idiopathic Interstitial Pneumonia [15]. Any discrepancy in the interpretation was solved by consensus. The fibrotic component, defined by reticular opacities and inflammation by ground-glass opacities, was graded according to the Goh [16] scores. We evaluated the agreement in the evaluation of the extent of pulmonary disease with the Goh between the two experts (intraclass correlation coefficient 0.68 (95% confidence interval (CI): 0.54–0.82)). The assessment of expert M.M. was applied in the analysis of the data; M.M. has a high intra-observer agreement (intraclass correlation coefficient 0.90 (95% CI: 0.84–0.94)).

### 2.4. Autoantibodies

The IgG anti-tRNA (anti-Jo1, anti-PL7, anti-PL12, anti-Ej, and anti-Oj) was measured using EUROIMMUN immunoblot strips (EUROLINE: Myositis Profile 3) according to the manufacturer’s instructions. This commercial line blot assay for myositis diagnosis was assessed on its diagnostic accuracy against RNA immunoprecipitation in a multicenter cohort of patients with Idiopathic inflammatory myopathies (IIM). The overall specificity of the line blot was 100% for anti-Jo1, anti-PL-7, and anti-PL-12 [17].

### 2.5. Cytokines’ Immunoassays

Serum cytokine levels of all patients were determined using a commercially available 18plex Human ProcartaPlex (Thermo Fisher Scientific, Waltham, MA, USA, cat. EPX180-12165-901) according to the manufacturer’s instructions. Interleukin (IL) -1β, IL-2, IL-4, IL-5, IL-6, IL-8, IL-9, IL-10, IL-13, IL-17A, IL-18, IL-21, IL-22, IL-23, IL-27, Tumor necrosis factor (TNF) α, Interferon (IFN)-γ, and Granulocyte-Macrophage Colony-Stimulating Factor (GM-CSF) were quantified. Multiplex assays were performed according to the manufacturer’s instructions. Samples were homogenized and adjusted for further quantification in the Luminex® LABScan 100 (Luminex Corp. Austin, TX, USA) system. The cytokine concentrations were calculated using the standard curve generated by the five-parameter logistic regression method. In samples where cytokines were undetectable, the values of the detection limit were used for analyses. The xPONENT 3.1 software (Luminex Corp. Austin, TX, USA) was used for data acquisition and data analysis.

### 2.6. Statistical Analysis

Categorical variables are described with frequencies and percentages, and numerical variables with a mean ± standard deviation (SD), or medians and interquartile range (IQR) according to the distribution of the variables. To compare baseline with follow-up cytokines serum levels, we used the paired *t*-test or Wilcoxon sign rank as appropriate. A comparison of baseline and follow-up cytokines levels was performed with the *t*-test or the Wilcoxon sign rank test as necessary, between patients with ILD progression with those with ILD improvement or stability. To compare baseline clinical and cytokine levels according to the anti-tRNA profile, the Kruskal–Wallis test or one-way analysis of variance (ANOVA) was used as appropriate; if a difference was found, a comparison between each group was made according to the Bonferroni correction. In the case of categorical variables, the exact Fisher test was used to evaluate if there were differences at baseline according to the anti-tRNA profile. Finally, Receiver Characteristic Operating (ROC) curves were elaborated to estimate the discriminatory capacity of cytokines being assessed, to differentiate between ILD progressors patients (PP) or no progressors patients (NPP). All analyses were two-sided, α was set at 5% unless otherwise specified. The statistical software Stata v. 14.2 was employed to perform all analyses.

## 3. Results

### 3.1. Patients Included

Thirty-nine ILD patients positive to anti-tRNA autoantibodies (10 anti-Jo1, 10 anti-Ej, 11 anti-PL12, and eight anti-PL7) were included. Autoantibodies were mutually exclusive, and 69.23% of the patients were positive to anti-Ro52. Most patients were females (71.7%) mean age 51.9 ± 9.87 years old and had a median of pulmonary symptoms before baseline evaluation of 12 months (IQR: 7–24 months). Only six patients (20%) had three or more Bohan and Peter’s criteria [18] to be classified as possible or definitive inflammatory myopathy; on the contrary, most patients fulfilled interstitial pneumonia with autoimmune features criteria [2] (30.77%). The most frequent HRCT finding was Organized Pneumonia (OP) pattern, followed by the Nonspecific Interstitial Pneumonia (NSIP) pattern and Usual Interstitial Pneumonia (UIP) pattern. Patients had a great extent of ground glass in HRCT scan according to the Goh score and a low extent of fibrosis in HRCT, rendering the Goh score (Table 1). The CK level at baseline had a median of 100 (62–384), the percentage of patients with arthritis was 72%, with fever 51.28% and 60% positives to Mechanic’s hand sign (Table 1). 

### 3.2. PFTs at Baseline and Follow-Up

The percentage baseline median of predicted FVC was 50 (41–74), and the percentage baseline median of predicted DL_CO_ was 45 (33–66). At six months after initiating medical treatment, the percentage median of predicted FVC was 68 (51–92), and the percentage median of predicted DL_CO_ was 61 (40–87). Both six months’ percentages of predicted were different when compared to baseline (*p* = 0.02 and *p* = 0.0001 for FVC and DL_CO,_ respectively). Three patients had ILD progression; most patients (26; 67%) had ILD improvement. The rest of the patients had lung disease stability. Table 2 shows the baseline comparison between those patients with ILD progression with those with ILD improvement or stability. Only CK baseline levels had a statistical difference, with lower values of CK in patients who had ILD progression (*p* = 0.01) (Table 2). On another hand, comparison of clinical features according to the anti-tRNA autoantibodies is shown at Table 3.

### 3.3. Serum Cytokines Quantification

Table 4 and Table 5 show the comparison of the serum concentration of cytokines at baseline and follow-up.

IL-6 was lower at baseline (median 1694.06 pg/mL, IQR 430.04–2313.54 pg/mL) compared to the levels at follow-up (median 2298.40 pg/mL, IQR 456.86–2358.95 pg/mL; Figure 1); and IL-22 was lower (median 1017.11 pg/mL, IQR 824.67–1058.23 pg/mL) at baseline compared to the levels at follow-up (median 1062.48 pg/mL, IQR 870.15–2262.52 pg/mL; Figure 2). Table 4 shows the comparison of cytokine levels among different antibodies at baseline. Only the serum levels of IL-27 showed statistically significant differences between patients anti-Jo1+ (median 453 pg/mL, IQR 447–469 pg/mL) and patients anti-PL7+ (median 474 pg/mL, IQR 458–483 pg/mL). Table 5 shows the same comparison with levels at follow-up. Although initially a probable difference in the levels of IL-1β was observed, the Bonferroni correction revealed that these differences were not significant.

On the other hand, when discriminating patients between PP or NPP at follow-up, the Th17 inflammatory cytokine profile showed statistically significant differences, being in all cases higher in those patients who showed ILD progression and a high probability of discrimination between both conditions calculated with ROC curves: IL-1β (NPP, median 222.97 pg/mL, IQR 92.20–223.11 pg/mL), (PP, median 673.63 pg/mL, IQR 440.12–907.15 pg/mL); IL-4 (NPP, median 694.95 pg/mL, IQR 382.41–786.11 pg/mL), (PP, median 2225.03 pg/mL, IQR 1890.60–2559.46 pg/mL); IL-6 (NNP, median 2298.40 pg/mL, IQR 442.47–2353.91 pg/mL), (PP, median 4351.61 pg/mL, IQR 2639.59–6063.62 pg/mL); IL-10 (NPP, median 70.84 pg/mL, IQR 62.53–82.81 pg/mL), (PP, median 467.58 pg/mL, IQR 221.22–713.94 pg/mL; Figure 1); IL-12 (NPP, median 304.86 pg/mL, IQR 281.53–339.77 pg/mL), (PP, median 589.53 pg/mL, IQR 473.74–705.31 pg/mL); IL-17A (NPP, median 188.22 pg/mL, IQR 125.96–286.52 pg/mL), (PP, median 490.13 pg/mL, IQR 285–620.30 pg/mL); IL-18 (NPP, median 1058.36 pg/mL, IQR 854.44–1438.75 pg/mL), (PP, median 4362.28 pg/mL, IQR 3102.60–5621.95 pg/mL); IL-22 (NPP, median 1053.26 pg/mL, IQR 860.82–1962.40 pg/mL), (PP, median 3195.38 pg/mL, IQR 3174–7358.54 pg/mL); GM-CSF (NPP, median 792.71 pg/mL, IQR 687.51–856 pg/mL), (PP, median 2637.53 pg/mL, IQR 1223.07–4052 pg/mL); TNF-α (NPP, median 405.89 pg/mL, IQR 353.36–408.68 pg/mL), (PP, median 1516.44 pg/mL, IQR 1408.71–1624.16 pg/mL; Figure 2).

### 3.4. Association between Cytokines

The correlations among cytokines IL-10 and IL-17A; IL-23 and IL-17A; and IL-23 and IL-10, was made in all patients that participated in the protocol using the levels detected in the follow-up. In all cases, the correlations were highly statistically significant (Figure 3).

## 4. Discussion

This study which was designed to evaluate the association between the cytokines’ concentrations and the pulmonary outcome of ILD patients positive to anti-tRNA autoantibodies, shows that the Th17 profile of immune response is associated with ILD progression. This report describes for the first time that the Th17 inflammatory profile could have a fundamental role in the pathophysiology of ILD associated with anti-tRNA and the possible role of the cytokines of the Th17 inflammatory profiles as biomarkers and therapeutic targets in anti-tRNA-ILD.

ASSD seems to have different clinical phenotypes, categorized by the types of anti-tRNA. These clinical phenotypes differ in prognosis and frequency of clinical manifestations; for example, patients anti-PL7+ and anti-PL12+ autoantibodies are associated with higher frequency and severity of ILD and higher mortality rate [19]. This information coincides with one of the most important findings in the current work, that is, only anti-PL7 and anti-PL12 positive patients had ILD progression, and these patients also had different clinical manifestations at baseline (including lower levels of CK), and more importantly, these patients had the higher levels of cytokines of the Th17 inflammatory profile at follow-up (Table 3). 

According to the results of this study, the baseline cytokines levels are consistent with a highly heterogeneous inflammatory response; however, at follow-up, only the patients with ILD progression showed higher levels of cytokines such as IL-1β, IL-4, IL-6, IL-10, IL-17A, IL-18, IL-22, GM-CSF, and TNF-α; these findings allow us to frame the immune response, such as the Th17 profile, and propose a pathophysiological pathway (Figure 4).

The discriminatory capacity of many of these cytokines, evaluated with the ROC methodology, suggests that these may have an essential role as biomarkers, for example, IL-6 had an area under a ROC of 0.98 at follow-up. Moreover, in the era of biological therapies and JAK/Stat inhibitors, these suggest that IL-6 and other cytokines may be an important therapeutic target in the treatment of these patients.

The results of follow-up cytokines levels, even with medical treatment, suggest that the inflammatory process is still active and is more severe in the PP; this is in concordance with observations that, although many patients may have a pulmonary improvement with medical treatment, these patients are at high risk of relapses of the pulmonary disease in the long term, as was described by Yamakawa et. al [20]. The results of this study justify the evaluation of medical treatments blocking the Th17 inflammatory profile, to reduce the high risk of relapse of lung disease in this group of patients.

There is evidence that the lung is a crucial organ for the onset and development of autoimmune diseases such as rheumatoid arthritis (RA) [21]. Infections, tissue injury, and metabolic disorders may cause inflammatory environments in the lungs, where cells react by producing cytokines [7,22]. Depending on the type of stimulus, the innate immunological mechanisms can be activated and provide a bridge for sensitization of the production of autoantibodies [23,24].

Under physiological conditions, the coordinated action of cells and molecules of the immune system results in a reduction of circulating antigens [25]; however, the persistence of the inflammatory stimulus, the immune response tends to stay and polarize [26]. In autoimmune diseases, the effects of IL-10 are only inhibitory; although, TGF-β can act synergistically with proinflammatory cytokines and even causes fibrotic lesions and promotes autoimmune pathologies [27]. It has been documented that the binding of TGF-β with IL-6 or IL-1-β induces differentiation of T-lymphocyte naive to Th17 [28]. Th17-lymphocytes are highly efficient producers of IL-21, IL-22, and especially IL-27; cytokines widely associated with the development of pathologies such as Systemic Lupus Erythematosus (SLE) [29], RA [28], and Multiple Sclerosis (MS) [30]. The mechanisms by which the Th17 immune response would favor the development of anti-tRNA-ILD are diverse; it has been recognized that the synergistic activity between IL-6 and TGF-β induces the transformation of regulatory T-lymphocytes (Treg) into Th17-lymphocytes, thus reducing the number of cells competent of counteracting the activity of self-reactive T- [31] and B-lymphocytes [32].

On the other hand, IL-6 establishes a bridge between the activation of T- and B-lymphocytes; IL-6 promotes the survival, activation, expansion, and maturation of B-lymphocytes and plasma cells, promoting the development of follicular T-lymphocytes [33], these cells located in the follicles of B-cells within the lymph nodes, promote the proliferation and class switching of B-lymphocytes through cytokines such as IL-4 [34], as previously described in autoimmune diseases such as SLE [35]. Further, IL-17A stimulates synoviocytes to induce IL-8 and CCL20, which recruits neutrophils and lymphocytes to the joint, respectively, and in fibroblasts triggers IL-6 secretion, which aggravates tissue inflammation. Finally, high levels of IL-17A have been implicated with tissular infiltrates composed primarily of CD8+ T-lymphocytes in autoimmune diseases [36], including ILD in idiopathic inflammatory myopathies [37], and particularly with severe complications of ASSD [38].

We recognize that the fact that only three of the patients had progression of ILD makes it difficult to compare with those patients without progression to follow-up; unfortunately, we are unable to present a clinically relevant predictor of disease severity at the baseline. However, the findings open the possibility of carrying out protocols in the future with specific searches in molecules related to the Th17 inflammatory profile.

Since the importance of the different cytokines in the development of autoimmune diseases, it has been proposed that the relationship established between them may indicate a trend in the development of the pathology, the relationship between IL-10 and IL-17A would indicate a limitation in the inflammatory phenomenon [39]. At the same time, the ratio of IL-23 and IL-17A would show the possibility of increased inflammation [28,40]. In order to explore these relationships, both balances were evaluated (Figure 3). In the analyzed patients, both relationships were statistically significant, therefore we could hypothesize that there is a profoundly positive effect in the control of the disease through treatment; however, the inflammatory process is still present chronically.

The detected levels of the different cytokines of Th17 profile, allow to integrate the anti-tRNA-ILD into the group of autoimmune diseases where this profile has an essential role in its progress. Furthermore, these results open the possibility of initiating studies on the molecules involved in this inflammatory pathway as biomarkers and therapeutic targets.

## 5. Highlights

Th17 inflammatory profile is associated with the progression of interstitial lung disease in patients with anti-tRNA-ILD.

Cytokines of the inflammatory Th17 profile show high discrimination capacity between ILD progressor and non-progressor patients.

The anti-PL7 and anti-PL12 autoantibodies had a different inflammatory phenomenon compared to the anti-Jo1 phenotype of ASSD and are associated with the progression of ILD and show higher levels of Th17 cytokines.

The Th17 inflammatory profile could have a fundamental role in the pathophysiology of ILD associated with anti-tRNA autoantibodies.

## Figures and Tables

**Figure 1 jcm-09-01356-f001:**
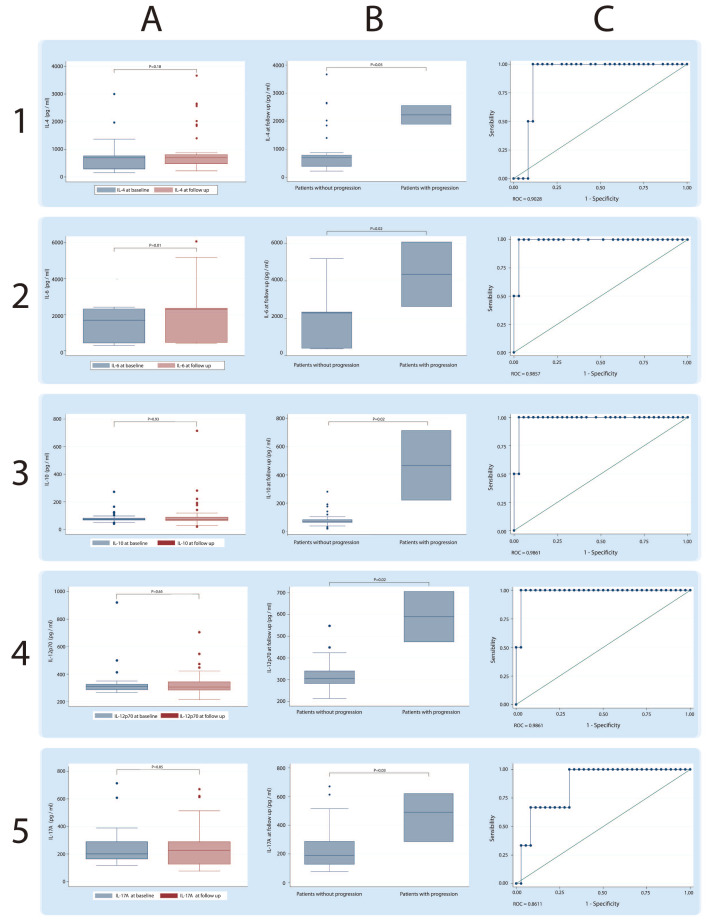
**Serum concentrations of cytokines IL-4, IL-6, IL-10, and IL-12P70 in patients positives for anti-synthetase syndrome (ASSD) autoantibodies.** Each row shows a particular cytokine. Column **A** shows the global comparison at baseline and the follow-up; Column **B** shows the comparison made between patients with progression and patients without progression of interstitial lung disease (ILD), and Column **C** shows the discrimination capacity of each cytokine calculated using ROC curves.

**Figure 2 jcm-09-01356-f002:**
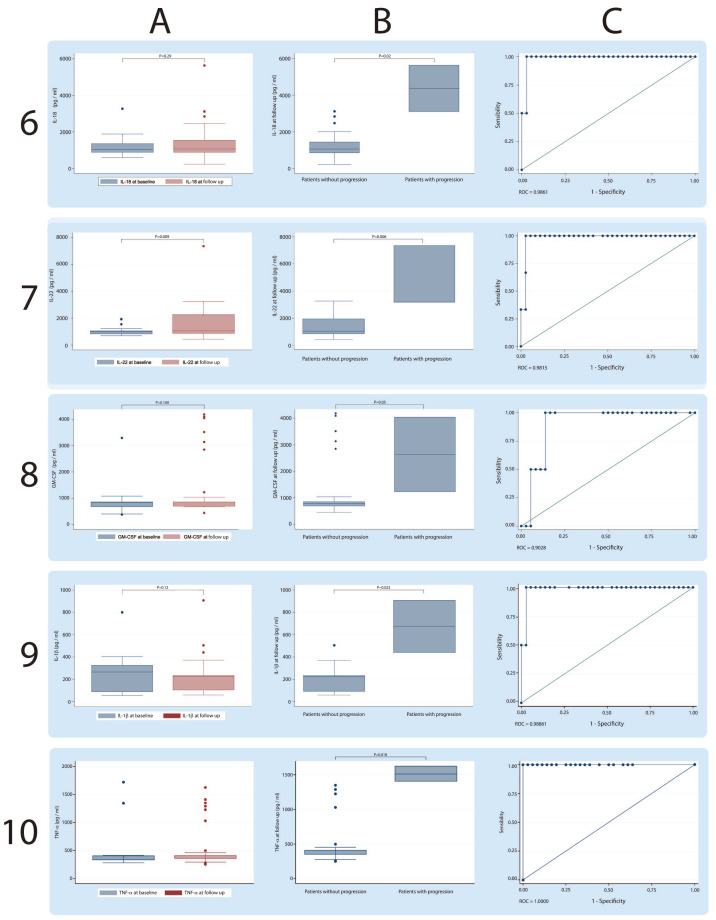
**Serum concentrations of cytokines IL-18, IL-22, GM-CSF and TNF-****α in patients positive for ASSD autoantibodies.** Each row shows a particular cytokine. Column **A** shows the global comparison at baseline and the follow-up; Column **B** shows the comparison made between patients with progression and patients without progression of ILD, and Column **C** shows the discrimination capacity of each cytokine calculated using ROC curves.

**Figure 3 jcm-09-01356-f003:**
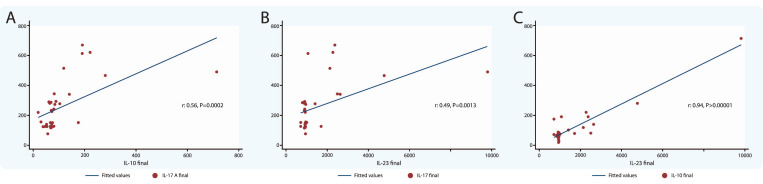
**Correlation between serum concentrations of cytokines.** (**A**) Correlation global concentrations at follow-up between IL-10 and IL-17A. (**B**) Correlation global concentrations at follow-up between IL-23 and IL-17. (**C**) Correlation global concentrations at follow-up between IL-23 and IL-10.

**Figure 4 jcm-09-01356-f004:**
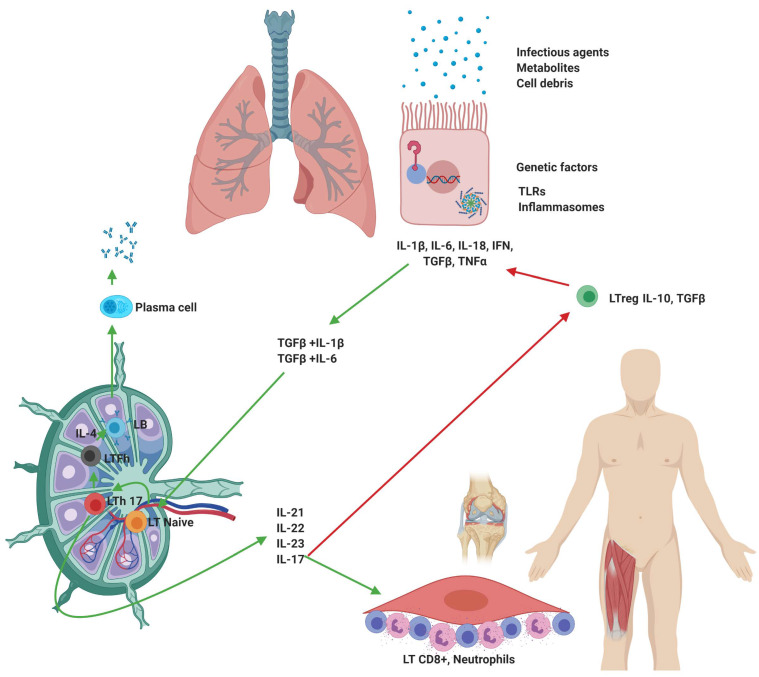
**Th17 inflammatory profile in the antisynthetase syndrome.** Proposed pathophysiological pathway with the integration of cytokines of the Th17 inflammatory profile (IL-1, IL-6, IL-17A, IL-18, IL-21, IL-22, IL-23 TGFβ, and TNF-α). The green arrows indicate stimulatory pathways and red arrows inhibitory stimuli.

**Table 1 jcm-09-01356-t001:** Description of the patients included in the study.

Variable	*n* = 39
Age at baseline, years old.	51.9 ± 9.87
Gender female: male	28 (71.7%):11 (28.3%)
Pulmonary symptoms before baseline evaluation	12 (7–24) months
Anti-tRNA autoantibody	
Jo1+ (%)	10 (25.64)
PL7+ (%)	8 (20.51)
PL12+ (%)	11 (28.20)
Ej+ (%)	10 (25.64)
Ro52+ (%)	27 (69.23%)
HRCT pattern	
OP	20 (51.28%)
NSIP	17 (43.59%)
UIP	2 (5.13%)
Lung disease extent in HRCT *	56 (40–62)
Ground glass extent in HRCT *	51.52 (34.4–57.04)
Fibrosis extent in HRCT *	5.6 (3.72–8.88)
Baseline % of predicted value of FVC (median, (IQR))	50 (41–74)
Baseline % of the predicted value of the DL_CO_	45 (33–66)
CK level at baseline	100 (62–384) min, max: 22–6318
Arthritis	28 (72%)
Fever	20 (51.28%)
Mechanic’s hand sign	23 (60%)
Raynaud’s phenomenon	13 (33%)
Proximal muscle weakness	28 (72%)
Modified Borg scale at rest	4 (2–5)
Modified Borg scale during activity	6 (5–9)

* According to the Goh score [16].

**Table 2 jcm-09-01356-t002:** Comparison interstitial lung disease (ILD) patients positive to anti-tRNA, with ILD progression, against subjects who evolved to improve ILD.

Variable	Patients with ILD Progression *n* = 3	Patients without ILD Progression *n* = 36	*p*-Value
Age at baseline assessment	52 (49–83)	51 (45.5–56)	0.42
Female gender	2 (66.6%)	34 (94.45%)	1.0
Pulmonary symptoms before baseline evaluation	12 (12–24)	10.5 (3.5–20)	0.33
Anti-tRNA autoantibody			
Jo1+	0	10 (27.77%)	
PL7+	1 (33.33%)	7 (19.44%)	
PL12+	2 (66.66%)	9 (25%)	
Ej+	0	10 (27.77%)	0.27
HRCT pattern			
OP	2 (66.66%)	18 (50%)	
NSIP	1 (33.33%)	16 (44.44%)	
UIP	0	2 (5.55%)	1.0
Lung disease extent in HRCT *	63 (56–70)	56 (40–62)	0.34
Ground glass extent in HRCT *	53.2 (50.4–56)	51.2 (34.4–57)	0.86
Fibrosis extent in HRCT *	9.8 (5.6–14)	4.96 (3.72–8.5)	0.26
Baseline % of predicted value of FVC (median, (IQR))	66 (36–77)	49 (41.5–74)	0.87
Baseline % of predicted value of the DL_CO_	55 (15–66)	44.5 (33.5–67)	0.95
CK level at baseline	42 (22–50)	105.5 (72.5–516.5)	0.01
Arthritis	2	1	1.0
Fever	1	2	1.0
Modified Borg scale at rest	4 (2.5–5)	3 (2–3)	0.21
Modified Borg scale during activity	6.5 (5–5.5)	5 (4–6)	0.17

* According to the Goh score [16].

**Table 3 jcm-09-01356-t003:** Comparison of clinical features, pulmonary function tests (PFTs) according to the anti-tRNA autoantibodies.

Variable	Anti-Jo1+ Patients *n* = 10	Anti-PL7+ Patients *n* = 8	Anti-PL12+ Patients *n* = 11	Anti-EJ+ Patients v10	*p*-Value
Age at baseline assessment	45 ± 9.5	62 ± 10.7	51.5 ± 5.6	50.9 ± 7.4	0.002 ^∃^
Female gender *	9 (90%)	7 (87.5%)	4 (45.5%)	9 (90%)	0.13
Pulmonary symptoms before baseline evaluation	7.5 (2–14)	15 (5–24)	12 (1–20)	12 (6–12)	0.87
Patients with ILD progression at follow-up.	0	1 (12.5%)	2 (18.18%)	0	0.27
Arthritis *	10 (100%)	5 (62%)	8 (73%)	5 (50%)	0.06
Mechanic’s hand sign *	8 (80%)	5 (62.5%)	7 (63.6%)	3 (30%)	0.16
Fever *	7 (70%)	3 (37.5%)	4 (36.4%)	6 (60%)	0.37
Raynaud’s phenomenon *	6 (60%)	3 (37.5%)	2 (19%)	2 (20%)	0.18
CK level at baseline ***	516 (107–2413)	61 (34–75.5)	100 (50–178)	240 (88–1652)	0.004
Anti-Ro52 positivity *	5 (50%)	5 (62.5%)	8 (72.7%)	8 (80%)	0.55
Baseline % of predicted value of FVC **	65 (34–76)	75 (59–86.5)	45 (42–64)	44 (40–60)	0.20
Baseline % of predictive value of DL**_CO_**	51 (38–80)	48 (29.5–61.5)	40 (33–55)	45.5 (32–73)	0.85
Patients with improvement in FVC (>10%) or in DL**_CO_** (>15%) *	6 (60%)	4(50%)	6 (54.5%)	10 (100%)	0.047
Patients with lung disease progression in FVC (>10%) or in DL**_CO_** (15%) *	0	1 (12.5%)	2 (19%)	0	0.27
Follow-up % of predicted value of FVC	74 (57–104)	66 (49–95.5)	58.5 (43–62)	71 (43–92)	0.35
Follow-up % of predicted value of DL**_CO_**	80 (55–119)	70 (37–89)	56 (40–57)	61 (35–87)	0.37

* Categorical variables are described with percentages. ** Medians (IQR). ^∃^ Anti-PL7+ patients are different from anti-Jo1+ and anti-Ej+ patients at baseline age evaluation, *p* < 0.001 and < 0.049, respectively, anti-PL7 tended be older compared to anti-PL12+ patients (*p* < 0.064). *** Anti-PL7 had statistically lower CK levels compared to Anti-Jo1+ (*p* < 0.0034), and anti-Ej (*p* < 0.009). Anti-PL12+ patients had lower baseline CK levels compared to anti-Jo1+ patients (*p* < 0.03).

**Table 4 jcm-09-01356-t004:** Baseline cytokine levels according to the anti -tRNA profile and in the complete cohort.

Variable	Anti-Jo1+ Patients *n*: 10	Anti-PL7+ Patients *n*: 8	Anti-PL12+ Patients *n*: 11	Anti-EJ+ Patients *n*: 10	*p*-Value	ILD anti tRNA Positive Patients *n*: 39
**IL-1β**	92 (75.3–310)	202 (65–327)	319 (107–335)	239 (105–318)	0.22	264 (88–324)
**IL-2**	262 (236–284)	280 (214–286)	278 (273–282)	279 (272–284)	0.82	278 (247–284)
**IL-4**	288 (277–624)	522 (261–761)	730 (446–786)	695 (516–744)	0.09	688 (283–758)
**IL-5**	408 (308–458)	354 (226–465)	453 (370–459)	453 (440–463)	0.79	452 (311–463)
**IL-6**	445 (423–2243)	1509 (437–2283)	2205 (415–2318)	1996 (440–2324)	0.67	1694 (438–2313)
**IL-9**	454 (383–495)	430 (389–524)	434 (412–480)	463 (450–565)	0.71	454 (412–495)
**IL-10**	70 (67–73)	73 (48–85)	74 (71–97)	76 (70–84)	0.31	73 (68–81)
**IL-12 p70**	321 (287–331)	317 (289–324)	302 (290–318)	305 (279–326)	0.88	311 (287–326)
**IL-13**	105 (95–124)	144 (108–189)	153 (117–167)	123 (106–190)	0.21	119 (104–174)
**IL-17A**	232 (167–290)	230 (146–290)	193 (151–290)	214 (176–289)	0.89	201 (164–289)
**IL-18**	908 (795–1408)	907 (814–1170)	1065 (1022–1292)	1037 (966–1441)	0.23	1022 (875–1353)
**IL-21**	283 (175–379)	349 (259–505)	340 (273–358)	345 (294–564)	0.64	346 (233–450)
**IL-22**	871 (740–952)	1026 (730–1064)	1027 (981–1058)	1017 (883–1048)	0.37	1017 (824–1058)
**IL-23**	898 (863–947)	917 (795–979)	937 (916–1003)	932 (903–1259)	0.28	927 (885–976)
**IL-27**	453 (447–469) *	474 (458–483)	474 (464–492)	483 (264–324) *	0.02	473 (458–485)
**INF-γ**	661 (624–724)	698 (607–950)	815 (633–938)	819 (668–856)	0.41	720 (644–860)
**GM-CSF**	856 (725–856)	808 (667–856)	843 (688–856)	686 (662–856)	0.77	843 (670–856)
**TNF-α**	346 (326–405)	365 (313–404)	405 (403–407)	403 (349–404)	0.07	403 (332–405)

The units of the serum cytokine concentrations were pg/mL in all cases. All values are expressed as medians (IQR). ***** Patients positive for anti-Jo1 and anti-EJ showed statistically significant differences after having performed the Bonferroni correction.

**Table 5 jcm-09-01356-t005:** Follow-up cytokine levels according to the anti -tRNA autoantibodies in the complete cohort.

Variable	Anti-Jo1+ Patients *n* = 10	Anti-PL7+ Patients *n* = 8	Anti-PL12+ Patients *n* = 11	Anti-EJ+ Patients *n* = 10	*p*-Value	ILD anti tRNA Positive Patients *n* = 39
**IL-1β**	92 (70–225)	235 (187–405)	224 (214–234)	223 (104–233)	0.05 *	264 (88–324)
**IL-2**	256 (223-272)	279 (238–479)	276 (264–561)	268 (266–273)	0.19	271 (249–288)
**IL-4**	594 (288–786)	754 (471–2595)	691 (660–803)	709 (528–744)	0.60	698 (472–803)
**IL-5**	357 (235–445)	443 (440–1722)	438 (428–476)	437 (313–442)	0.09	438 (337–447)
**IL-6**	650 (435–2308)	2339 (2127–5185)	2298 (1709–2349)	2313 (430–2359)	0.38	2298 (456–2358)
**IL-9**	441 (357–558)	512 (3893–1871)	551 (536–622)	505 (398–548)	0.39	534 (398–593)
**IL-10**	64 (61–75)	129 (73–235)	69 (66–84)	72 (68–80)	0.12	71 (63–89)
**IL-12 p70**	322 (296–328)	308 (278–435)	298 (275–351)	296 (284–342)	0.90	305 (283–344)
**IL-13**	111 (97–143)	123 (99–518)	102 (72–211)	306 (283–345)	0.76	109 (97–172)
**IL-17A**	279 (220–286)	288 (152–477)	225 (140–287)	125 (124–143)	0.13	225 (126–290)
**IL-18**	1128 (672–1535)	1231 (845–2653)	1108 (1039–1504)	1050 (831–1260)	0.82	1064 (878–1535)
**IL-21**	526 (131–959)	583 (341–4368)	327 (281–495)	331 (297–740)	0.18	345 (297–765)
**IL-22**	1408 (729–2025)	1298 (1032–2698)	1058 (786–3174)	1045 (949–2017)	0.81	1062 (870–2262)
**IL-23**	923 (900–945)	1033 (915–3631)	932 (922–956)	929 (910–939)	0.44	932 (908–1065)
**IL-27**	475 (457–489)	477 (462–1742)	481 (475–2359)	481 (470–6790)	0.60	479 (461–1650)
**INF-γ**	786 (631–843)	1042 (760–1391)	843 (691–991)	713 (669–738)	0.14	815 (669–932)
**GM-CSF**	855 (679–856)	2084 (765–3786)	710 (691–856)	716 (689–856)	0.28	855 (689–856)
**TNF-α**	380 (309–407)	406 (377–1259)	405 (403–410)	408 (404–409)	0.33	406 (359–409)

The units of the serum cytokine concentrations were pg/mL in all cases. All values are expressed as medians (IQR). * After the correction of Bonferroni, no significant differences were observed in any group in the serum concentration of IL-1β.

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
