# Peer review of "Anti-Aminoacyl Transfer-RNA-Synthetases (Anti-tRNA) Autoantibodies Associated with Interstitial Lung Disease: Pulmonary Disease Progression has a Persistent Elevation of the Th17 Cytokine Profile"

_jcm, 2020, doi:10.3390/jcm9051356_

Round 1

Reviewer 1 Report

I had the pleasure of reviewing the manuscript by Ramos-Martinez et al. titled “Interstitial lung disease positive to antisynthetase syndrome autoantibodies: Pulmonary disease progression has a persistent elevation of the Th17 cytokine profile”.

I congratulate the authors of this paper, as they performed a very interesting and well-designed study.

The subject is of great relevance in the area of interstitial lung disease, both for prognostic and therapeutic purposes.

I think this manuscript needs moderate amendments before publication.

My main concerns are:

TEXT:

  • English language needs revision.
  • ABSTRACT and INTRODUCTION:
  1. Your paper starts describing ASSD. The first reference of this sentence refers to IPAF. As you specified in “patients included”, only 6 patients satisfied the classification criteria for idiopathic inflammatory myopathy, while most patients fulfilled the classification for IPAF. Therefore, you should describe ASSD as well as IPAF and the possible differential diagnosis for anti-tRNA Abs positivity. References need revision (reference 1 refers to IPAF, please add a reference for ASSD).
  2. The gap ofknowledge is not emphasized. What’s the need of clinicians and the relevance for this research? (e.g. lack of biomarkers and therapeutic guidelines in ASSD and anti-tRNA?). Please, could you clarify it?
  3. Th17 cytokine profile is not mentioned at all. Could you explain the role of these cytokines in ASSD or anti-tRNA? What does literature say on this topic? (another gap of knowledge?)
  • MATERIAL and METHODS:
  1. Did you use a dyspnea scale (for example: MRC, Borg)?
  • RESULTS:
  1. The second sentence of the section “3.3 quantification of serum cytokines” is not clear (page 6, lines 190-192).
  • TABLES: please, add the reference for Goh score in table 1 and 2.
  • FIGURES:
  1. In legends to figures 1 and 2 you describe column. What do rows refer to?
  2. Figure 4: could you specify what is the Th17 inflammatory profile in the figure? For example: “Th17 inflammatory profile: IL1b, IL6, ...”.
  • DISCUSSION:
  1. You compared the serum cytokines of PP with NPP. However, progressive patients (PP) were only 3. You should consider this limitation.
  2. The Th17 inflammatory profile was higher in PP only in the follow-up. Thus, it is not properly to be considered as a predictive biomarker or a risk factor. You should consider this limitation.
  3. Did your patients experience acute exacerbations of ILD?
  4. The topic of acute exacerbation in ILD could be of great interest for your research. Could you add some speculations on this topic in the discussion? (For example: Papiris SA, Tomos IP, Karakatsani A, Spathis A, Korbila I, Analitis A, Kolilekas L, Kagouridis K, Loukides S, Karakitsos P, Manali ED. High levels of IL-6 and IL-8 characterize early-on idiopathic pulmonary fibrosis acute exacerbations. Cytokine. 2018 Feb;102:168-172. doi: 10.1016/j.cyto.2017.08.019.).

REFERENCES:

  • In the text, reference numbers should be placed in square brackets [ ], for example [1], [1–3] or [1,3].
  • In the section “References”, the references are not list with the Journal format. For Journal Articles, they should be described as follows: Author 1, A.B.; Author 2, C.D. Title of the article.Abbreviated Journal Name YearVolume, page range.
  • Reference number 5: You rightly say that ASSD treatment is based on the use of CS and a wide variety of immunosuppressors. However, you cite a paper that described the use of Cyclosporine. Moreover, your therapeutic regimen did not include Cyclosporin, but methotrexate and leflunomide. You should cite a more general article on this topic (for example: Vacchi C, et al. Therapeutic Options for the Treatment of Interstitial Lung Disease Related to Connective Tissue Diseases. A Narrative Review. J Clin Med. 2020 Feb 3;9(2).).
  • Some references are lacking. For example:

-page 3, line 102: “…established criteria for IPF [?].”

-page 4, line 165: “…Bohan and Peter’s criteria…[?]”

- page 4, line 166: “…IPAF criteria… [Fischer 2015]”

  • What does reference 19 refer to? You describe the most frequent HRCT pattern of your study; thus, reference is probably a mistake…

I would like to congratulate the authors on their work. This was a very interesting manuscript andI hope to read more of their work in the future.

Thank you.

Best regards.

Author Response

The authors want to thank the reviewer for their time and dedication, the observations made helped us improve the text presented.

 The suggestions were addressed as follows:

TEXT:

  • English language needs revision.

R: Thank you for your kind comments. We have performed an English language revision through the whole manuscript by a native speaker.

  • ABSTRACT and INTRODUCTION:
  1. Your paper starts describing ASSD. The first reference of this sentence refers to IPAF. As you specified in “patients included”, only 6 patients satisfied the classification criteria for idiopathic inflammatory myopathy, while most patients fulfilled the classification for IPAF. Therefore, you should describe ASSD as well as IPAF and the possible differential diagnosis for anti-tRNA Abs positivity. References need revision (reference 1 refers to IPAF, please add a reference for ASSD).

R:  Thank you very much for the comment. We made several modifications to the manuscript to answer this concern. Also, the references have been revised.

  1. The gap Knowledge is not emphasized. What’s the need of clinicians and the relevance for this research? (e.g. lack of biomarkers and therapeutic guidelines in ASSD and anti-tRNA?). Please, could you clarify it?

R: We have modified the introduction section to address this comment.

  1. Th17 cytokine profile is not mentioned at all. Could you explain the role of these cytokines in ASSD or anti-tRNA? What does literature say on this topic? (another gap of knowledge?)

R: Now we have modified the current version of the manuscript to include the Th17 cytokine profile.

  • MATERIAL and METHODS:
  1. Did you use a dyspnea scale (for example: MRC, Borg)?

R: We evaluated the modified Borg dyspnea score; this information is now included in tables 1 and 2.

  • RESULTS:
  1. The second sentence of the section “3.3 quantification of serum cytokines” is not clear (page 6, lines 190-192).

R: Now, we have corrected it.

  • TABLES: please, add the reference for Goh score in table 1 and 2
  1. Now is included, as you requested. 
  • FIGURES:
  1. In legends to figures 1 and 2 you describe column. What do rows refer to?

R: Each row shows a particular cytokine analyzed.

  1. Figure 4: could you specify what is the Th17 inflammatory profile in the figure? For example: “Th17 inflammatory profile: IL1b, IL6, ...”.

R: Now is included.

  • DISCUSSION:
  1. You compared the serum cytokines of PP with NPP. However, progressive patients (PP) were only 3. You should consider this limitation.

R: We agree with the comment, this is a limitation of our study, and it is recognized in the discussion section.

  1. The Th17 inflammatory profile was higher in PP only in the follow-up. Thus, it is not properly to be considered as a predictive biomarker or a risk factor. You should consider this limitation.
  2. We recognize in the text that the number of patients with progression is a limitation of work; however, the results are significant and guide us to more specific future explorations.
  3. Did your patients experience acute exacerbations of ILD?

R: No patients included in the study experienced an acute exacerbation of ILD during follow up.

  1. The topic of acute exacerbation in ILD could be of great interest for your research. Could you add some speculations on this topic in the discussion? (For example: Papiris SA, Tomos IP, Karakatsani A, Spathis A, Korbila I, Analitis A, Kolilekas L, Kagouridis K, Loukides S, Karakitsos P, Manali ED. High levels of IL-6 and IL-8 characterize early-on idiopathic pulmonary fibrosis acute exacerbations. Cytokine. 2018 Feb;102:168-172. doi: 10.1016/j.cyto.2017.08.019.).

R: We thank the reviewer for his comment, and of course acute exacerbations are of great interest in the prognosis of ILD patients, nevertheless, as no patient in this cohort had an acute exacerbation of ILD, it is hard to include in the discussion section some theories on this topic.

REFERENCES:

  • In the text, reference numbers should be placed in square brackets [ ], for example [1], [1–3] or [1,3].
  • In the section “References”, the references are not list with the Journal format. For Journal Articles, they should be described as follows: Author 1, A.B.; Author 2, C.D. Title of the article.Abbreviated Journal NameYearVolume, page range.
  • Reference number 5: You rightly say that ASSD treatment is based on the use of CS and a wide variety of immunosuppressors. However, you cite a paper that described the use of Cyclosporine. Moreover, your therapeutic regimen did not include Cyclosporin, but methotrexate and leflunomide. You should cite a more general article on this topic (for example: Vacchi C, et al. Therapeutic Options for the Treatment of Interstitial Lung Disease Related to Connective Tissue Diseases. A Narrative Review. J Clin Med. 2020 Feb 3;9(2).).
  • Some references are lacking. For example:

-page 3, line 102: “…established criteria for IPF [?].”

-page 4, line 165: “…Bohan and Peter’s criteria…[?]”

- page 4, line 166: “…IPAF criteria… [Fischer 2015]”

  • What does reference 19 refer to? You describe the most frequent HRCT pattern of your study; thus, reference is probably a mistake…
  1. Each one of the references was reviewed and as a whole the format was corrected

 We appreciate the reviewers comment.

Reviewer 2 Report

Authors compared cytokine level using samples obtained from patients with or without ILD progression after six months of treatment. It is better to compare using samples obtained at baseline too.

Authors should descript and compare cytokine data of each patients  both at baseline and after six months of treatment. 

There is concern that the number of ILD progression patients is too less to discuss.

Also, there are many miss-descriptions which have to be improved in this article.

line; 178 CVF ⇒ FVC

line; 179 (26(67%) ⇒ (26(67%))

Figure 1. IL-1, IL-4, IL-6, IL-10, AND IL-12P70 ⇒ IL-4, IL-6, IL-10, IL12P70, IL-17

Figure 1. 1B IL6 ⇒ IL-1

Figure 1. at follow line ⇒ at follow up

Figure 1. ex) IL-1 at followup (pg/ml) ⇒ IL-1 (pg/ml)

Figure 2. IL-17A, IL-18, IL-22, GM-CSF, AND TNF-α ⇒ IL-18, IL-22, GM-CSF, IL-1β, AND TNF-α

Author Response

The authors want to thank the reviewer for their time and dedication, the observations made helped us improve the text presented.

 The suggestions were addressed as follows:

Authors compared cytokine level using samples obtained from patients with or without ILD progression after six months of treatment. It is better to compare using samples obtained at baseline too.

Authors should descript and compare cytokine data of each patients  both at baseline and after six months of treatment.

 R: We improved the presentation of the data for a better appreciation of the comparison made at basaline and at follow-up, and at follow-up between progressors and non-progressors

There is concern that the number of ILD progression patients is too less to discuss.

  1. We recognize in the text that the number of patients with progression is a limitation of work; however, the results are significant and guide us to more specific future explorations.

Also, there are many miss-descriptions which have to be improved in this article.

line; 178 CVF ⇒ FVC

line; 179 (26(67%) ⇒ (26(67%))

Figure 1. IL-1, IL-4, IL-6, IL-10, AND IL-12P70 ⇒ IL-4, IL-6, IL-10, IL12P70, IL-17

Figure 1. 1B IL6 ⇒ IL-1

Figure 1. at follow line ⇒ at follow up

Figure 1. ex) IL-1 at followup (pg/ml) ⇒ IL-1 (pg/ml)

Figure 2. IL-17A, IL-18, IL-22, GM-CSF, AND TNF-α ⇒ IL-18, IL-22, GM-CSF, IL-1β, AND TNF-α

  1. All the misdescriptions were attended to improve the presentation of results.

We appreciate the reviewers comment.

Round 2

Reviewer 2 Report

If we could predict therapy response from baseline cytokine levels, that is interesting. So I commented as follows in report 1. Authors compared cytokine level using samples obtained from patients with or without ILD progression after six months of treatment. It is better to compare using samples obtained at baseline too.
Authors should descript and compare cytokine data of each patients  both at baseline and after six months of treatment.
But authors could not improve as demanded level.

And authors concluded that "there is an association between the cytokines of the Th17 inflammatory profile and the ASSD progression.". But other T cells and  macrophages also can secrete cytokines which are elevated in this study. Authors does not show enough evidence to support that conclusions.

Author Response

We appreciate each of the comments made by the reviewer, we have addressed the comments to improve our work.

Reviewer: If we could predict therapy response from baseline cytokine levels, that is interesting. So I commented as follows in report 1. Authors compared cytokine level using samples obtained from patients with or without ILD progression after six months of treatment. It is better to compare using samples obtained at baseline too. Authors should descript and compare cytokine data of each patients both at baseline and after six months of treatment. But authors could not improve as demanded level.

Response: We agree with the reviewer, that a predictor of disease severity and bad prognosis at baseline evaluation would be clinically relevant, Nevertheless, there were no baseline cytokine differences in no ILD progressors VS ILD progressors, so unfortunately, we are unable to  present a clinically relevant predictor of disease severity at the baseline. Nevertheless, our results are clinically relevant due to the description that a Th17 inflammatory response has an important role in the progression of ILD in anti RNAt positive patients. This knowledge may have relevance in the design of treatment schemes in this group of patients. We have recognized this limitation of our study in the new version of the manuscript.

Reviewer:   And authors concluded that "there is an association between the cytokines of the Th17 inflammatory profile and the ASSD progression.". But other T cells and macrophages also can secrete cytokines which are elevated in this study. Authors does not show enough evidence to support that conclusion

Response: The evaluation of inflammatory profiles includes cytokines that stimulate lymphocyte differentiation, cytokines that are produced in response to lymphocyte activity and properly those produced by differentiated lymphocytes. In our manuscript, we conclude that the Th17 inflammatory profile is associated with ASSD progression, due that cytokines of this profile are observed, but not necessarily produced only by lymphocytes; we cannot deny the participation of different cell lines in a coordinated way. We have clarified this point in the new version of the discussion section. We thank the reviewer for his comment.